# Phosphite Application Alleviates *Pythophthora infestans* by Modulation of Photosynthetic and Physio-Biochemical Metabolites in Potato Leaves

**DOI:** 10.3390/pathogens9030170

**Published:** 2020-02-28

**Authors:** Mohammad Aqa Mohammadi, Xiaoyun Han, Zhizhong Zhang, Yupei Xi, Mohammadreza Boorboori, Gefu Wang-Pruski

**Affiliations:** 1Joint FAFU-Dalhousie Lab, College of Horticulture, Fujian Agriculture and Forestry University, Fuzhou 350002, Chinazeada2001@163.com (Z.Z.);; 2Department of Horticulture, College of Agriculture, Alberoni University, Kapisa 1254, Afghanistan; 3College of Life Science, Fujian Agriculture and Forestry University, Fuzhou 350002, China; m.boorboori@yahoo.com; 4Department of Plant, Food and Environmental Sciences, Faculty of Agriculture, Dalhousie University, Truro, NS B3H 4R2, Canada

**Keywords:** Potassium phosphite, potato, late blight, *Phytophtora infestans*, chlorophyll, oxidative enzymes, antioxidant compounds

## Abstract

Potato late blight (*Phytophtora infestans*) is among the most severely damaging diseases of potato (*Solanum tuberusom* L.) worldwide, causing serious damages in potato leaves and tubers. In the present study, the effects of potassium phosphite (KPhi) applications on photosynthetic parameters, enzymatic and non-enzymatic antioxidant properties, hydrogen peroxide (H_2_O_2_) and malondialdehyde (MDA), total protein and total carbohydrate of potato leaves challenged with *P. infestans* pathogen were investigated. Potato leaves were sprayed five times with KPhi (0.5%) during the growing season prior to inoculation with *P. infestans*. The potato leaves were artificially infected by the LC06-44 pathogen isolate. The leaves were sampled at 0, 24, 48, 72 and 96 h after the infection for evaluations. *P. infestans* infection reduced chlorophyll (Chl) pigments contents, chlorophyll fluorescence, carotenoid (Car) and anthocyanin contents and increased the accumulation of H_2_O_2_ and MDA. Meanwhile, our result showed that KPhi treatment alleviated adverse effect of late blight in potato leaves. KPhi application also increased plant tolerance to the pathogen with improved photosynthetic parameters Chl *a, b*, total Chl, Car, and anthocyanin compare to controls. Moreover, the increased oxidative enzymes activity of superoxide dismutase (SOD), peroxidase (POD), catalase (CAT) and ascorbate peroxidase (APx), and non-enzymatic substances such as phenolics, flavonoids and proline were found in KPhi treated plants, compared to untreated plants after inoculation. In addition, KPhi application followed by *P. infestans* infection also decreased the content of H_2_O_2_ and MDA, but increased the total protein and total carbohydrate contents in potato leaves. The consequence of current research indicated that KPhi played a vital role in pathogen tolerance, protecting the functions of photosynthetic apparatus by improved oxidative levels and physio-biochemical compounds in potato leaves.

## 1. Introduction

Late blight caused by Oomycete *Phytophthora infestans* is a main disease of potato worldwide, resulting in a significant crop loss annually [1]. Fungicides are broadly applied to control this pathogen by preventing pathogen’s invasion, inhibiting pathogens growth or targeting and destroying pathogen reproduction [2]. Some fungicides also have indirect effect by activating plant immune system to boost plant resistance against biotic and abiotic stresses [3,4].

It has been described that several chemicals can induce plant defense responses. These compounds are known as resistance inducers [5]. Phosphites (HPO_3_^2-^, Phi), alkaline salts of phosphoric acid (H_3_PO_4_), are known to control Oomycete pathogens by stimulating plant defense responses [6]. Phi can directly inhibit oxidative phosphorylation in Oomycete metabolism [7], and indirectly trigger plant defense mechanism, ultimately inhibit pathogenesis [8]. Phi plays a vital role as a fungicide or biostimulator [9,10]. In addition, Phi application increased expression of pathogenesis-related (PR) genes and increased soluble protein accumulations in Arabidopsis, potato and tomato [5,11]. Potassium phosphite (KPhi) was shown to increase systemic acquired resistance (SAR) signalling pathways in different types of plants to control diseases induced by Oomycetes, such as Phytophthora spp. [3], Pythium spp. [12] and Pseudoperonospora spp. [13].

The interplay among plant and pathogen conducts to some changes in physiological and bi-chemical metabolism in the host plant. For instance, photosynthetic reactions, respiration rate, carbon assimilation can be significantly stimulated during pathogen invasion [14]. Pathogens can decrease photosynthetic activities or interrupt biochemical pathways in several plant species [15]. Researchers have reported a significant reduction in photosynthesis, decreasing the activity of photosystem II (PS II) and reducing quantum production between many commonly occurring plant reactions after being inoculated with the pathogen [16]. Biotic stresses damage PS II and other parts of the electron transport chain, leading to significant reduction in the chlorophyll (Chl) pigments and transport of photosynthesis electrons [17]. Some of the essential features and benefits of immune system have been identified. Research has been focused on exploiting these findings to decrease the damage caused by diseases [18]. One report explored that during biotic and abiotic stresses, high concentrations of reactive oxygen species (ROS) were produced, which could result in loss of various intro-cellular macro-molecules [19,20]. The equilibrium between ROS that results from the damage to the PS and activities of antioxidant enzymes can distinguish plants safety [21]. Preliminary reports showed that high level of resistance of pre-treated plants with KPhi against pathogen stresses was mainly associated with high concentration of defense enzymes in host plant tissues after environmental stresses [22].

Plant antioxidant defenses, oxidative enzymes productions, have expanded with aerobic alteration to balance oxidative damage caused by ROS. Protective enzymes included catalase (CAT), ascorbate peroxidase (APx), and superoxide dismutase (SOD), and various molecules including glutathione, proline, ascorbate and carotenoid (Car) have non-enzymatic protection functions [19,20,23,24,25,26]. Other reports [27,28] have shown that Phi usage decreased assembly of H_2_O_2_ in plant fresh tissues. Similar to other biological tensions, fungal infection produced extra free radicals, for example, hydroxyl radicals, H_2_O_2_ and superoxide [29]. It has been well described that after activation of antioxidant enzymes, plant tolerance to pathogen challenges obviously increased [30].

Earlier investigations have documented that changes in activity of ROS scavenging enzymes can also be a crucial step in activating plant defense against plant pathogens [31]. The activity of the antioxidant enzyme system has been noted to limit the release of oxidative proteins and enable cells to resist the inoculation of *Aphanomyces euteiches* and *S. sclerotiorum* in plant cells [32,33,34]. Among several common metabolites, phenolics and flavonoids compounds have protective activity with free radicals and antiseptic properties in plants [35]. The accumulation of proline occurred in response to many biotic and abiotic stressed tensions, for example fungal infection [34]. Proline acted as a strong ROS holder and prevented the death of cells that was induced by ROS [36]. Anthocyanin is usually associated with increased activity of phenylalanine ammonium lyase (PAL) and has a biological role such as defense feature against various pathogens, free radicals and antioxidants activities [37]. Carbohydrates play an essential role in responding to different stresses, in interaction with plant microbes, and exert as signal molecules to regulate genes expression, such as PR genes [38].

In present study, we investigated the five times application of KPhi (0.5%) on potato leaves to alleviate adverse effects of *P. infestans* in potato leaves. Therefore, the effect of KPhi was studied as resistance inducers against *P. infestans* in potato. In addition, the effects of KPhi on photosynthetic parameters including Chl (*a*, *b*), total Chl content and Chl fluorescence in potato after *P. infestans* infection were studied. Furthermore, four key antioxidants enzyme activity (SOD, POD, CAT and APx), four non-enzymatic compounds including phenolics, flavonoids, proline and anthocyanin, and ROS associated metabolites such as H_2_O_2_ and MDA, total soluble protein and total soluble carbohydrate were also measured after *P. infestans* infection. This study provided a comprehensive view to KPhi mode of action involved after late blight infection in potato plant. It will lay the foundation for researches of further molecular mechanism regarding Phi mode of action in plants.

## 2. Results

### 2.1. Effect of KPhi on Leaf Protection

The potato leaves were strongly protected from pathogen infection by using five times application of KPhi after emergence (Figure 1). Control leaves from 48 h showed small lesions and the highest level of lesion area occurred at 96 h (Figure 1A). The leaves treated with KPhi showed small lesion size at 96 h after inoculation. This result indicated that KPhi could protect potato leaves from developing disease symptoms, and such a protection in KPhi leaves was observed with about 50% decrease in disease severity compared to non-treated leaves (Figure 1B).

### 2.2. Effect of KPhi on Photosynthetic Pigments

To assess effect of exogenous KPhi (0.5%) treatment on potato leaves after late blight infection, changes in Chl *a*,*b*, total Chl, Car, and Chl fluorescence were estimated. *P. infestans* infection led to physiological changes in photosynthetic pigments in control, KPhi treated and *P. infestans* infected leaves. Figure 2A demonstrated the changes in Chl *a* in potato leaves after treated by KPhi (KPhi), after pathogen infection (CPi) or pathogen infection after KPhi treatment (KPhiPi) in comparison to the control (C). As seen, Chl content gradually increased within 96 h in control samples. KPhi treatment also increased Chl content within 96 h when compared to control. After leaves were infected, Chl *a* content drastically decreased within 96 h. However, when treated with KPhi (KPhiPi), such a decrease is less significant (Figure 2A). When looking into Chl *b* (Figure 2B), similar patterns were found. In this case, KPhi treated leaves contained more Chl *b* than control; and Chl *b* content in KPhiPi is much less decreased compared with CPi. This comparison is more accurate than Chl *a* as the initial Chl *b* values from both CPi and KPhi and CPi and KPhiPi are the same. It also explained that leaves from KPhiPi treated were greener and healthier looking than that of CPi (data not shown).

In case of the total Chl, changes were similar to Chl *a* and Chl *b*, with gradually increased in control samples (C) as compared to KPhi at all time points. Similarly, reduced amount of photosynthesis in pathogen infected samples had induced the reduction of the amount of leaf total Chl. Our results indicated that total Chl was reduced in control samples (CPi) within 96 h after inoculation in potato leaves, when compared to KPhi treated plants, however such a decrease was much less in KPhi treated samples after inoculation (Figure 2C).

Likewise, the highest amounts of Car contents were found at 96 h in KPhi treated plants compared to control samples (Figure 2D). The Car content progressively increased in both control and KPhi treated samples, although in KPhi treated samples, the Car content was higher compared to control at 96 h. Inoculation by *P. infestans* lowered the content of this pigment in KPhi treated and in untreated plants (Figure 2D). As shown in Figure 2D, the Car content is dramatically reduced in control samples after inoculation (CPi), when compared to KPhiPi which the decrease was slightly higher within 96 h after inoculation. Plants treated with KPhi maintained content of Chl *a*, Chl *b*, total Chl, as well as Car under pathogen stress when compared to control (Figure 2A–D). 

Figure 2E,F showed the Chl fluorescence, as an indicator of change in photosynthetic pigments. It was shown that when plastaquinone electroplastic (Qa) fully oxidized (Fo value) and maximum quantum production of PS II (Fm/Fv value) were decreased in control and KPhi treated plants after infection (Figure 2E,F). Fo value in control leaves (C) showed minor increases at all time points, while the treated leaves significantly increased within 96 h compared to control leaves. Meanwhile, *P. infestans* damaged the Chl fluorescence (Fo value) mostly in untreated leaves 96 h after inoculation. KPhi treatment had reduced adverse effect of pathogen infection on Fo value. Fo value in KPhiPi were reduced less compared to untreated leaves following the infection. The result indicated that inoculation by pathogen had a negative effect on normal function of PS II of photosynthetic system in potato leaves. The Fv/Fm value changes in control and KPhi treated leaves were similar to Fo value (Figure 2F). But in KPhi treated leaves, Fv/Fm values increased with the highest peak recorded at 96 h. *P. infestans* inoculation gradually decreased Fv/Fm value (CPi) within all time points, but the values in KPhiPi showed less reduction (Figure 2E). Fv/Fm represents the quantum yield of PS II photochemistry in dark adopted state, the decrease of Fv/Fm means that photo-inhibition of photosynthesis of potato leaves increased after inoculation. Taken together, our results showed protective effects of KPhi on Chl fluorescence where mainly Fo and Fv/Fm values were alleviated by application of 0.5% KPhi on potato leaves (Figure 2E,F). Moreover KPhi application caused a significant up-regulation of PS II activity and reduced the damage of dissipate ability by pathogen invasion.

### 2.3. Effect of KPhi on Enzymatic Antioxidant Compounds

KPhi application significantly increased activity of SOD, POD, CAT and APx, within 96 h after inoculation with the pathogen. As shown in Figure 3A, SOD activity was gradually increased in control plants from 0 to 96 h. SOD activity in KPhi samples did not showed significant difference at all time points. The plants without KPhi treatment after infection had lowest activity at 0 h and then rapidly increased to 96 h after inoculation. When CPi is compared with KPhiPi samples, the SOD activities were sharply increased in KPhiPi samples after 48 h of infection (Figure 3A). It means that SOD activities were significantly promoted by KPhi treatment after inoculation by pathogen. 

Moreover, an increased in POD enzymatic activity in potato leaf tissues was observed in control (C) after inoculation with pathogen (CPi) (Figure 3B). The lowest activity was seen in all treatments at 0 h. POD activity slowly increased from 0 to 96 h in control leaves, when it compared with KPhi treatment. There was no obvious difference at all treatment times, except at 96 h when a significant increase was found in KPhi treated samples. After pathogen inoculation, POD activity gradually increased in control samples (CPi), but increase in its activity was more significant in KPhiPi samples, as demonstrated after 24 h inoculation in KPhi treated plants (Figure 3B). 

The CAT activity changed at all time points. As shown in Figure 4C, CAT activity increased gradually during the trial period in control and KPhi treatment at all time points from 0 to 96 h. No significant differences were observed between control and KPhi treated samples at all time points. CAT activity was slightly increased and then decreased sharply after inoculation (CPi), but what is interesting that CAT activity in KPhiPi samples significantly increased after 24 h infection by the pathogen (Figure 3C).

APx activity increased rapidly in the control treatment and in all three treatment groups (Figure 3D). Again, the highest APx activity in plants was observed in KPhiPi samples with significant increase started at 24 h after inoculation and continued. Overall, KPhi-applied leaves exhibited more APx activity than control plants over time. Our result showed that exogenous KPhi pursued healing of oxidative stress caused by pathogen, which is related to up-regulation of antioxidant enzymes activity.

### 2.4. Effect of KPhi on Non-enzymatic Antioxidants Compounds

To find out role of KPhi on non-enzymatic accumulation, we measured phenolics, flavonoids, proline and anthocyanin compounds in potato leaves. Phenolics contents gradually increased in all samples during experiment (Figure 4A). The control plants showed gradual increased amount of phenolics at 0, 24, 48, 72 and 96 h. The same observations were made in KPhi treatment (KPhi). The control samples after inoculation (CPi) had the lowest phenolics content within 96 h. The highest amount of phenolics was observed in KPhiPi treated plants at all time points after inoculation (Figure 4A). In the present experiment, KPhi treated plants produced more phenolics content in comparison to control plants at all time points.

The amount of flavonoids did not change significantly during 0 to 96 h when control samples were compared with KPhi, except at 96 h where flavonoids content was significantly higher in KPhi samples (Figure 4B). After the pathogen infection, the flavonoids content was significantly increased until reached the highest level at 96 h (Figure 4B). This means that KPhi can increase flavonoids content in plant leaves after pathogen attack.

Investigation on proline in control leaves showed that the level from 0 to 96 h did not obviously changed (Figure 4C). When compared with other three treatment groups, the control samples had the lowest proline content. Again, as shown in Figure 4C, KPhi treatment (both KPhi and KPhiPi) increased proline contents at all time points, but the samples that were treated with KPhiPi) and then infected with the pathogen(showed the highest levels of proline in all time points (Figure 4C). This result indicated that KPhi could induce the proline content in plant leaves, and such an induction is more significant after leaves were infected by pathogen. 

KPhi also changed anthocyanin contents as shown in Figure 4D. Compared to control, the KPhi treated leaves showed high anthocyanin contents. Pathogenic infection (CPi) changed anthocyanin contents with the lowest amount recorded at 96 h when compared to treated leaves, but KPhiPi samples showed visible increase in anthocyanin contents at 0 to 96 h (Figure 4D). Nevertheless, overall changes in anthocyanin content were not dramatic when compared with many other parameters, such as phenolics and proline.

### 2.5. Effect of KPhi on ROS Compounds

The effect of KPhi on H_2_O_2_ and MDA accumulation in potato leaves after inoculation or non-inoculation with pathogen was also investigated. As shown in Figure 5A, the control samples alone had a gradual increase in H_2_O_2_ accumulation. Similar pattern was observed in KPhi applied samples (KPhi). Meanwhile, H_2_O_2_ content significantly increased in control plants after pathogen infection (CPi) at all time points. To our surprise, after pathogenic infections in plants treated with KPhi (KPhiPi), H_2_O_2_ level was increased slightly, then decreased at 72 and 96 h (Figure 5A). This result is particularly interesting, because it suggests that KPhi could reduce the H_2_O_2_ levels in leaf cells, in order to protect them from oxidation.

Regarding MDA content, again there was no significant difference between control and KPhi samples. The pathogen infection significantly increased MDA content in leaves (CPi), but when infected, the KPhi treated leaves (KPhiPi) showed significantly decreased MDA contents from 48 h (Figure 5B). This results further demonstrated that concentration patterns of MDA (Figure 6B) in the KPhiPi leaves were similar to those of H_2_O_2_ (Figure 5A). Overall, our research suggests that KPhi sufficiently alleviate adverse effect of late blight pathogen by modulation oxidative enzymes associated with ROS production in potato leaves.

### 2.6. Effects of KPhi on Total Protein and Total Carbohydrate Compounds

In our experiment, total soluble proteins were at lower levels in non-infected leaves (C) at all time points (Figure 6B). Significant increases happened in KPhi treatment at 96 h. In control plants after infection (CPi), the amount of total soluble proteins sharply increased at 0 to 24 h and rapidly raised from 48 to 96 h. While, the samples treated with KPhi (KPhiPi) increased even more dramatically, especially after 48 h (Figure 6B). This data is interesting because it seems the KPhi causes different responses in plants with and without the pathogen inoculation and more increase in total soluble proteins content in KPhiPi samples are worth further investigating. 

As shown in Figure 6A, the total soluble carbohydrates contents in control plants (C) and control infected plants (CPi) were not increased dramatically among sampling times. When compared to control (C) with KPhi, significant differences were found. The soluble carbohydrates contents were increased further in KPhi treated samples after infection (KPhiPi) (Figure 6A). The highlight in this data indicated that when treated by KPhi, the soluble carbohydrates content is higher, in both non-infected and infected leaf samples.

## 3. Discussion

This study investigated photosynthetic parameters changes, enzymatic and non-enzymatic activities due to KPhi application in potato leaves with and without infection by *P. infectans*. It is indicated, in our study, that pre-treatment of KPhi significantly decreased disease symptoms after infection with late blight pathogen, significant protection was observed in KPhi treated leaves (Figure 1A,B). This result is consistent with the previous research investigated by [39] that analyzed different concentrations of Phi on protection against late blight in potato leaves. Their study exhibited that late blight invasion on potato leaves was highly controlled, when Phi was used alone or combined with chlorothalonil against *P. infestans*. As well as [40] reported that Phi application had a rapid transient effect on the transcriptome, had quick response 3 h after treatment and protection was observed throughout all time points tested. 

Photosynthesis is a primary physiological process to enhance plant’s life through the use of light energy and synthesis of organic compounds in plants [41]. The efficiency of photosynthesis relies on content of Chl, therefore, ROS may cause a notable decrease in the amount of Chls in plant due to its delicate nature [42,43]. Therefore, survival, growth, and production of plants depend mainly on enhanced Chl contents. Our result exhibited that Chl concentrations in potato leaves were significantly reduced by pathogen infection, but KPhi enhanced Chl contents. Thus, application of KPhi increased the amount of Chl *a, b*, total Chl, Car, and Chl fluorescence (Figure 2). It was presented that this increase is perhaps due to notable induction of pigment formation to get higher photosynthetic efficiency with comprehensive resistance, which is associated with a decrease in amount of photophosphorylation that normally occurs after inoculation [44]. As seen in Figure 2A–D, plants treated with KPhi had higher Chl concentration than the untreated plants, which suggests that KPhi have a protective effect on Chl against pathogen in potato leaves. At the same time, a decrease in activation of Chl degrading enzymes was found in previous studies using KPhi to improve plants photosynthesis under different condition [42,45].

In addition, Chl fluorescence decomposition is a fast, effective, and adaptive path to detect photosynthesis in stressed plants [46]. The Chl fluorescence parameters (Fo and Fv/Fm) plays a vital role in energy conversion and is a good indicator of the pathogenesis in plants [47]. Our results showed that Chl fluorescence level, when Qa fully oxidized (Fo), reduced the maximum quantum yield of PS II (Fv/Fm) under inoculated potato leaves (Figure 2E,F). Similarly, a research explored that KPhi can enhance photochemical efficacy of the PSII [48].

In this study, antioxidant enzymatic activities were increased in pre-treated potato leaves with KPhi after infection with *P. infestans*, in comparison to control leaves. Comparable to present findings, our previous research showed that three times foliar application of KPhi (1%) to potato leaves protect tubers after inoculation by late blight pathogen resulted in elicitation of pathogenesis related protein enzymes, antioxidant enzymes activity and biochemical compound defense reactions. As well as these parameters are comparable with other research findings [29] in that KPhi increased growth and activities of antioxidant enzymes after infection with *Pythium ultimum* in cucumber plants. Likewise, SOD, CAT, APx, MDA and H_2_O_2_ increased in wheat after inoculated by *Pyricularia oryzae* [49].

Enzymes and biochemicals, for instance phenolicss, flavonoids, proline, and anthocyanin are produced by plants to defend against pathogens [50]. The production of phenolics belonged to stress response of potato, and our results exhibited a significant increase in phenolics contents in potato leaves treated with KPhi in comparison to control. The involvements of phytoalexin and phenolics in response to Phi were documented in various studies [51]. Flavonoids, as ROS, are damaged by tracing radicals as a leading contributor to potato and tomato plant in the face of environmental degradation [23,52]. In current research, we found that total concentration of flavonoids in potato leaves were increased in KPhi treated samples. In our current study, proline accumulation in KPhi treated leaves of potato was much higher under pathogen stress conditions in comparison to contents of control plants (Figure 4C). This may be due to increased proline synthesis, or increase in other amino acids that can alter proline [53]. In addition, [40] observed that Phi treatment activated stress responses and triggered significant changes in transcripts related to primary metabolism and secretome proteins predominantly associated with cell wall processes after 48 h. Their finding is in agreement with the current work which highlighted the response being activated at 48 h. 

When a plant is infected with pathogens, plant cell membrane integrity was destroyed, and membrane peroxidation was increased with production of ROS for example POD and H_2_O_2_ and increased levels of MDA. H_2_O_2_ is most active, toxic and destructive ROS and much stable compare to others ROS [54]. Expansion of free radicals and cellular losses eliminate the balance between formation and concentration of ROS. Results from earlier studies have shown that fungal invasion causes ROS production and reduces the balance between ROS production and detoxification [55,56]. Our findings showed that pathogen infection had increased H_2_O_2_ production in potato leaves, but this phenomenon is significantly mitigated by using KPhi in infected leaves (Figure 6A). We find out that *P. infestans* challenges significantly increased assembly of MDA (Figure 6B). Even though, the KPhi treatment and then pathogen inoculation significantly reduced the level of pathogen-mediated MDA (Figure 5B). Meanwhile, our results is in agreement with other studies [57]. H_2_O_2_ and MDA reduction indicates that KPhi application have protective effect on oxidative damages due to pathogen infection. A study reported by [58] on powdery mildew from the rubber tree indicated that MDA levels were increased in the infected plants. It is the first report that explored role of KPhi application mitigating ROS production after late blight infection in potato.

Our study showed that total protein and total carbohydrates increased in KPhi treated and then inoculated leaves in all samples during the times tested. However, the amount of carbohydrates in the control plants after inoculation with pathogen did not increased compared to KPhi, which may be due to malfunctioning of photosynthetic systems, which leads to a decrease in the production of carbohydrates and starches. Report [59] showed a positive correlation between carbohydrate and host resistance. Carbohydrates have been considered for their potential role as resistance inducers [60]. The perception and metabolism of sugars for microorganisms are limited by low level of penetration through cuticle [61]. Likewise, in our research KPhi increased carbohydrate content after pathogen inoculation in potato leaves (Figure 6B).

In this study, we showed that pre-application of KPhi is effective before pathogenic inoculation. This phenomenon suggests that KPhi increased the resistance level and activated plant defense responses before inoculation of *P. infestans*. Also, changes in activities of related enzymes were also studied. The findings of this study exhibited that use of KPhi in potato plants has enhanced levels of Chl *a, b*, total Chl, Car and Chl fluorescence. While the control plants had decreased Chl contents and Chl fluorescence. The damages caused by pathogen attack associated with ROS production, MDA and H_2_O_2_ accumulation decreased with KPhi dosage. Additionally, KPhi increased the level of antioxidant enzymes activity, phenolics, flavonoids, and anthocyanin which were considered as resistant biochemicals to *P. infestans*. Therefore, this study can be considered as new approaches to produce safe food and protect the environment where late blight is a limiting factor for potato production worldwide. In the end, a model for Phi-induced response against *P. infestans* before and after infection was proposed based on the data from this study, [40] and [62] (Figure 7). It explains the molecular basis to highlight Phi mode of action in potatoes against *P. infestans.* Indeed, Phi can be used a priming agent to trigger a wide range of plant defense functions against the pathogen. Recent work done by Wang-Pruski’s group also indicated that Phi may be affective to other pathogens, including some Verticillium species [63]. Therefore, further study on Phi’s mechanism may lead to its broader applications in agricultural production.

## 4. Materials and Methods

### 4.1. Experiment Location and P. infestans Culture

Chinese potato (*Solanum tubrusum* L.) Minshu No. 3 cultivar was used in our study. Minshu No. 3 was provided by Institute of Vegetables and Flowers, Chinese Academy of Agricultural Science. This cultivar is less susceptible to late blight. The Cv. Minshu No. 3 was grown in a greenhouse belongs to College of Horticulture at Fujian Agricultural & Forestry University (FAFU), Fuzhou city (latitude 26^0^, 5′, 16″ N, longitude 19^0^, 14′, 6″ E, altitude 42.09 m), in China from January 1, 2018 to April 20, 2018. Slices of seed tubers (~60 g weight possessing 3-4 eyes) were cut from tubers harvested three months earlier and planted in 5-liter plastic bags the substratrate contained mixture of vermiculite, peat and perlite (1:3:1). The temperature in greenhouse during growing season varied from 13 to 20 °C and daylight cycle were 10–14 h. The potato plants were watered with a hand sprinkler. Each experiment contained three replications, each replication contained 12 plants.

The *P. infestans* (Pi) (Mont.) LCO6-44 isolate locally taken from Fuqing city of Fujian province by personnel of College of Plant Protection at FAFU and was identified. The isolate cultured was kept at 18 °C and 90% relative humidity. After 7 days, mycelia were collected in sterile distilled water and stimulated to release sporangia by incubation at 4 °C for 6 h. After filtration through four layers of cheesecloth, the sporangial suspension was observed under an optical microscope for quantification before use as inoculums. The concentration of sporangia was adjusted to 4x10^4^ sporangia/mL using a hematocytometer, according to the published method [64].

### 4.2. KPhi Stock Solution, Preparation, Treatment and Foliage Assessment

Preparation of KPhi stock solution was described in details at our previous published method [64]. The phosphorous acid crystals were partially neutralized with potassium hydroxide (by continuous mixing of H_2_PO_3_ and KOH solution) and adjusted to pH 6.5. KPhi was applied on plants as leaf spray at a rate of (0.5 g/100 mL), when the seedlings were in 5-6 true-leave stage, 0.5% KPhi was applied, at value of 15 mL/plant (3 L/ha) using a hand sprayer. The applied dose was 0.5% (v/v) biweekly for five times in growing season. The control (C) plants were sprayed with water. Three days after last foliar application of KPhi, a fully expanded healthy leaf from each of control or KPhi treated leaves were harvested. In total, four leaves from four different plants (C or KPhi) per replicate were detached and wrapped in aluminum foil until inoculation. A total of three replications were made for each experiment. The entire sampling procedure was illustrated in Figure 8. The detached leaves were inoculated by spraying 500 μL of sporangial suspension (40,000 sporangia/mL) on the centre of the adaxial side of each leaf. The inoculated leaves were placed in Petri dishes with humid filter paper and incubated in a growth chamber in darkness at 15 °C for 24 h. After this 24 h period, leaf incubation continued in a growth chamber at 15 °C and 12 h day length. Disease progress on control and KPhi treated leaves was observed from 0 to 96 h post inoculation. Three independent samples of fresh leaves from each treatment group were pooled, weighed, homogenized to a fine powder according to the protocols for further investigation. The disease severity assayed by visual estimation of leaf area showing late blight lesion symptoms as described in our published methods [64]. The disease severity was recorded based on scale from 1 to 10, where 1 = no lesions, 2 = a few circles, 3 = up to 5%, 4 = 5–10%, 5 = 10–25%, 6 = 25–50%, 7 = 50–75%, 8 = 75–85%, 9 = 85–95% and 10 = 95–100% of leaf surface with late blight symptoms.

### 4.3. Photosynthetic Parameter Measurement

Fresh leaf samples (0.1 g) were grounded in mortar and pestle with 6 mL of 80% acetone. The samples were then centrifuged at 12,000 rpm for 20 min at 4 °C. The supernatant was used to measure the content of Chl *a, b,* total Chl and Car from potato leaves. The contents of Chl *a, b* and Car were measured by absorption in spectrophotometer at 663 nm, 645 nm and 470 nm. Then Chl *a, b*, total Chl and Car were calculated according to below equations [65].

Chl *a* (mg/g FW) = [12.7 (A_663_) − 2.69 (A_645_)] * V / W * 1000


Chl *b* (mg/g FW) = [22.9 (A_645_) − 4.68 (A_663_)] * V / W * 1000


Total Chl (mg/g FW) = Chl *a* + Chl *b*

Car (mg/g FW) = [1000(A_470_ − 2.05(Chl *a*) − 114.8/245]


In above relations, V is total volume of extracted sample, W is fresh weight and A is light absorption. 

Fluorescence of Chl was measured using factory performance analysis (PEA, Hansatech, Norfolk, UK). Estimation was made on plant leaves kept for 20 min in the dark before data recorded. Fluorescence was produced with a saturated PPFD of 3000 μmol m^−2^.s^−1^ get by an array of 6 LEDs (up to 650 nm). The minimum fluorescent level is determined when plastaquinone electroplastic (Qa) completely oxidized (Fo value) and maximum quantum production of PS II (Fv/Fm ratio) determined in fully expanded leaves. The Fv/Fm value was calculated as (Fm-Fo)/ Fm [66].

### 4.4. Measurement of Enzymatic Antioxidant Compounds

To extract the enzymes, 1 g of fresh leaf samples were homogenized in 2.5 mL sodium phosphate buffer (BPS) with pH 7.8 (Na_2_HPO_4_. 12H_2_O (16.385 g) + NaH_2_PO_4_. 2H_2_O (0.663 g) in sterilized water and made to a final volume of 1000 mL). The homogenate was centrifuged at 12,000 *g* for 20 min at 4℃ and supernatant was collected and kept at 4℃ for enzyme estimation. The supernatant was used to measure the enzymatic activities of SOD, POD, CAT, and APx. The antioxidant enzymatic activities assayed as described below.

To evaluate SOD activity, the published method with minor modification [67] was adapted. Reaction solution containing 20 μM/L riboflavin, 130 mM/L methionine, 100 µM/L EDTA-Na_2_ and 75 μM/L of nitro blue tetrazolium (NBT) was prepared. The 2.725 mL of the reaction mixture was delivered to small glass tubes, followed by addition 0.025 mL of enzyme extract and 0.25 mL H_2_O. The mixture was exposed for 15 min in an aluminum foil box containing 20 watt fluorescent tubes. A control tube, in which the sample was replaced with 2.75 mL of the reaction solution, was run in parallel at 560 nm. 

To asses POD activity, the guaiacol was used as a substrate and the analysis was based on the published method [67]. Increasing the absorption of guaiacol oxidation by H_2_O_2_ was recorded at 420 nm in a spectrophotometer. The reaction solution contained 50 mM PBS, 1.5% guaicol, and 300 mM H_2_O_2_. For enzyme activity, 2.7 mL BPS, 100 μL guaiacol, 100 μL H_2_O_2_ and 500 μL crude extract were combined and measured at room temperature. Control tube contained 2.8 mL BPS, 100 μL guaiacol and 100 μL H_2_O_2_.

The CAT was determined by measuring the disappearance of H_2_O_2_ at 240 nm [67]. The reagent mixture contained 50 mM sodium phosphate buffer (same as in case of grinding) and 300 mM H_2_O_2_ and 100 μL of extract. For enzyme activity, 2.8 ml BPS, 100 μL of the enzyme, 100 μL of H_2_O_2_ were taken, and the absorption was measured at 240 nm. The blank was prepared using the same solution without the enzyme extract.

APx was also evaluated by described method [67]. The reagent solution contained 100% dilute 50 mM phosphate buffer, 300 mM H_2_O_2_ and 7.5 mM ascorbic acid (ASA). For measuring the enzyme activity, 2.75 mL of BPS, 100 μL ASA, 100 μL H_2_O_2_ and 100 μL of enzyme extract were added in a tube. Distilled water was used as blank. The absorption decrease was recorded at 290 nm and the time was set 0–60 s.

### 4.5. Measurement of Non-enzymatic Antioxidant Compounds

To measure total phenolics content, fresh leaves sample (0.2 g) was grounded with 0.6 mL of water and 0.2 mL of folin-ciolcalteu previously diluted with water (1:1, V/V). After 5 min, 1 mL of sodium carbonate solution was saturated to the mixture and the volume of the reaction was adjusted to 3 mL with distilled water and the reaction solution was kept for 30 min in the dark and then centrifuged and absorbance of the samples at 765 nm was recorded in a spectrophotometer. Using a standard curve of gallic acid, the results were expressed as mg of gallic acid per gram of fresh weight (mg/FW) according to the method described by [68]. 

The total flavonoids concentration was estimated using fresh leaves (0.2 g) with 6 mL of 80% ethanol that was homogenized. Ethanol extract was centrifuged for 20 min at 4 °C at 12,000 rpm and supernatant was used to measure flavonoids. To determine total amount of flavonoids, a calorimetric method of aluminum chloride was applied and quercetin was applied to produce a standard calibration curve. To construct quercetin solution, 5.0 mg of quercetin was dissolved in 1 mL methanol. Then, a standard 0.6 mL quercetin solution was separately mixed with 0.6 mL aluminum chloride 2%. The solution was incubated for 60 min at room temperature. Absorption at 420 nm was measured in a spectrophotometer. The amount of flavonoids was calculated as mg/ FW [69].

Proline concentration was measured according to previous published methods [70]. To estimate proline, 0.2 g of fresh leaves sample grounded with 3% sulfosalicylic acid. The reaction mixture contained, acid ninhydrin, gallic acetic acid and toluene. Absorption was performed on a spectrophotometer at 520 nm and toluene was used as a blank. Proline content was recorded from a standard curve with standard solution L-proline. The outcomes were expressed in mg of L-proline g/FW.

### 4.6. H_2_O_2_ and MDA Determinations

Fresh leaves sample (0.1 g) was taken and homogenized in liquid nitrogen with 2 mL 0.1% (w/v) trichloroacetic acid (TCA) to detect H_2_O_2_. Homogenates centrifuged at 12,000 rpm for 15 min at 4 °C and supernatant (0.5 mL) was mixed with 10 mL potassium phosphate buffer (pH 7.0) and 1 mg potassium iodide solution (1 mL), and mixed for 5 min. The absorption of oxidation product formed in a spectrophotometer at 390 nm was estimated [71]. The concentration of H_2_O_2_ was calculated from the standard curve with known concentrations of H_2_O_2_ and expressed as fresh weight of μmol H_2_O_2_/g FW leaves. Malondialdehyde (MDA) content was determined by the TBA test (TBA), which determines MDA as final product of lipid peroxidation used to measure lipid peroxidation in leaves. MDA was measured according to the published method [72].

### 4.7. Total Protein Determination

Total soluble protein was determined according to previous method by [73]. Fresh leaf samples (0.2 g) were homogenized with 0.05 g of polyvinyl polypyrolidone (PVPP) and 0.05 M Tris buffer. Homogenate was centrifuged at 14,000 rpm for 20 min at 4 °C. Followed; 0.1 mL of supernatant was slowly mixed with 3 mL of Bradford reagent. The absorbance was measured at 595 nm (vs. blank) in a spectrophotometer after 5 min of mixing. Protein concentration was estimated using serum albumin as standard solution.

### 4.8. Total Carbohydrate Determination

To determine carbohydrate content, 0.2 g fresh leaf tissue in 5 mL of ethanol extract was homogenized at 80 °C for 15 min. The extracts were evaporated under vacuum at 70 °C under ethanol evaporation. To eliminate Chl pigments, extracts were mixed with chloroform and centrifuged for 5 min. The total carbohydrate were determined using a published method [74]. The absorption was recorded at 630 nm.

### 4.9. Statistical Analysis

All the experiments were designed in a complete randomized block design (CRBD), with three biological replicates. The data were subjected to significant variances and differences using one-way ANOVA of SPSS (version 22, SPSS, Chicago). Differences between means were detected, when using Tukey test (P < 0.05) and graphs generated by WPS Office Excel 2016.

## Figures and Tables

**Figure 1 pathogens-09-00170-f001:**
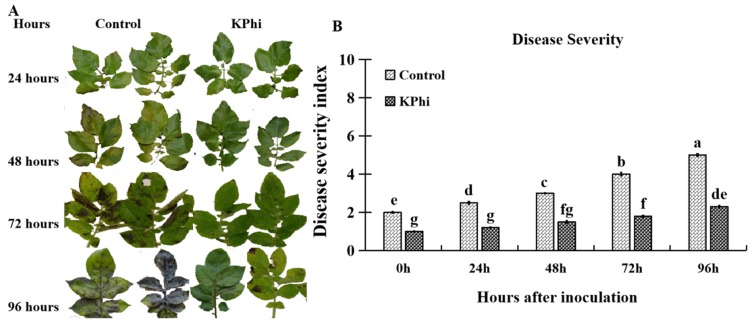
Disease severity after leaves was infected by *P. infestans*. (**A**). Phenotypic changes of potato leaves infected with *P. infestans* LC06-44 strain. The leaves were sprayed with 0.5% KPhi five times as treatment, and the leaves were not sprayed with KPhi as control. Two leaves from each of the control and KPhi treated plants were presented. Disease symptoms were recorded at 0, 24, 48, 72 and 96 h time points after infection. (**B**). Leaves disease severity was calculated from 0 to 96 h after inoculation. Means not sharing a common letter within a graph were significantly different at *p* < 0.05 according to the Tukey’s test.

**Figure 2 pathogens-09-00170-f002:**
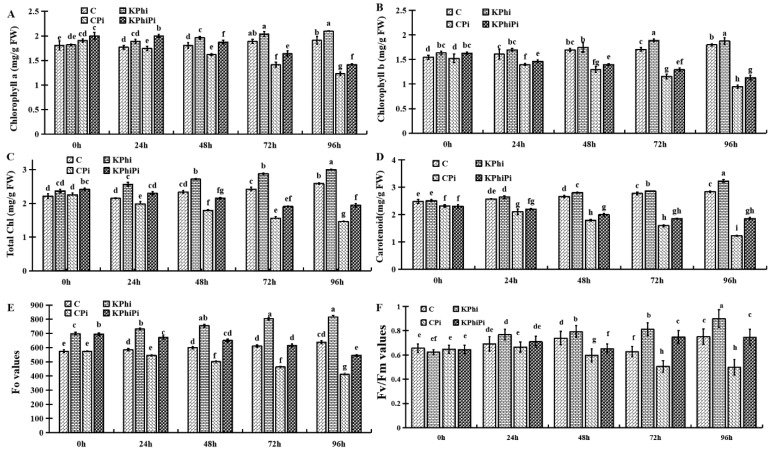
Photosynthetic changes in potato leaves. Chl *a* (**A**), Chl *b* (**B**), total Chl (**C**), Car (**D**) and Chl fluorescence, Fo value (**E**) and Fv/Fm value (**F**), C (control), KPhi (leaves treated with KPhi), CPi (control leaves inoculated by *P. infestans)*, KPhiPi (leaves inoculated three days after last KPhi application). Means not sharing a common letter within the same letters in the graph are significantly different at *p* < 0.05 according to Tukey’s test.

**Figure 3 pathogens-09-00170-f003:**
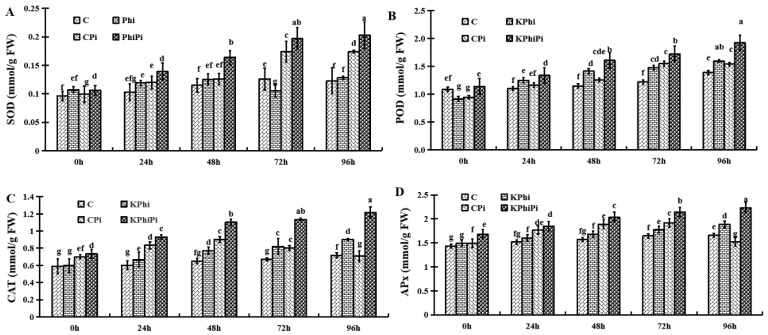
Antioxidants enzymatic activities in potato leaves, SOD (**A**), POD (**B**), CAT (**C**) and APx (**D**) in C (control), KPhi (leaves treated with KPhi), CPi (control leaves inoculated by *P. infestans*), KPhiPi (leaves inoculated three days after KPhi application). Means not sharing a common letter within the same letters in the graph are significantly different at *p* < 0.05 according to Tukey’s test.

**Figure 4 pathogens-09-00170-f004:**
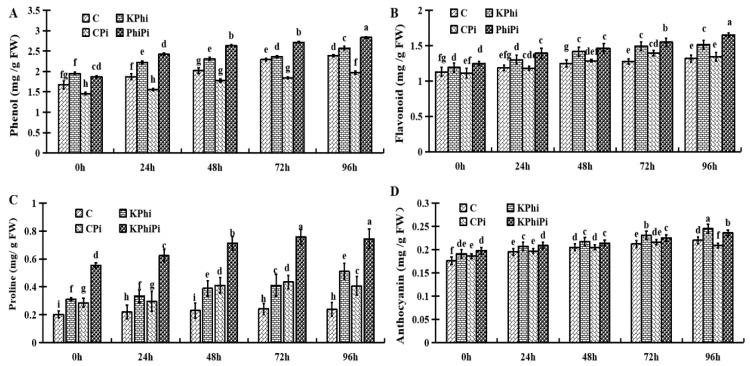
Changes of non-enzymatic and anthocyanin accumulation in potato leaves, phenolics (**A**), flavonoids (**B**), proline (**C**) and anthocyanin (**D**) in C (control), KPhi (leaves treated with KPhi), CPi (control leaves inoculated by *P. infestans*), KPhiPi (leaves inoculated three days after KPhi application). Means not sharing a common letter within the same letters in the graph are significantly different at *p* < 0.05 according to Tukey’s test.

**Figure 5 pathogens-09-00170-f005:**
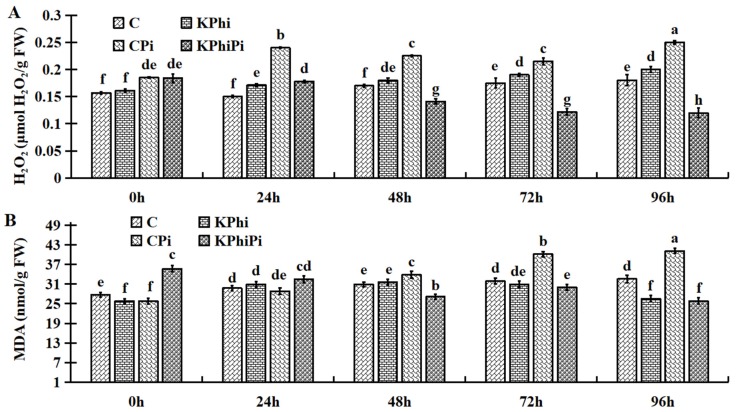
Changes of H_2_O_2_ (**A**) and MDA (**B**) in C (control), KPhi (leaves treated with KPhi), CPi (control leaves inoculated by *P. infestans*), KPhiPi (leaves inoculated three days after KPhi application). Means not sharing a common letter within the same letters in the graph are significantly different at *p* < 0.05 according to Tukey’s test.

**Figure 6 pathogens-09-00170-f006:**
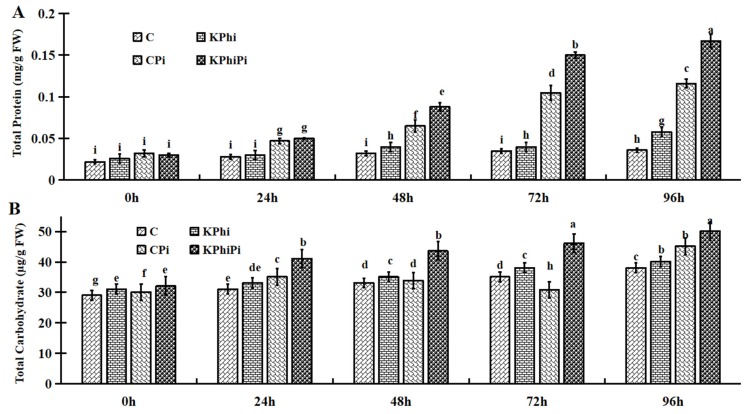
Changes of total protein content (**A**) and total carbohydrate (**B**) in C (control), KPhi (leaves treated with KPhi), CPi (control leaves inoculated by *P. infestans*), KPhiPi (leaves inoculated three days after KPhi application). Means not sharing a common letter within the same letters in the graph are significantly different at *p* < 0.05 according to Tukey’s test.

**Figure 7 pathogens-09-00170-f007:**
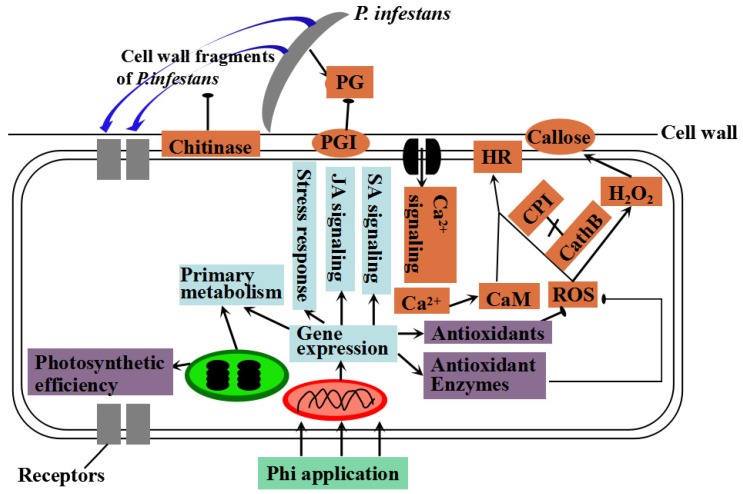
Proposed model for Phi-induced response against *P. infestans* before and after infection. Before infection, increased ROS and calmodulins (CaM) involved in Ca^2+^ signalling induced the hypersensitive response (HR). However, interactions between cysteine protease cathepsin B (Cath B) and cysteine protease inhibitors (CPI) blocked the activation of the HR. After infection, chitinase as the member of pathogenesis-related (PR) proteins was produced to degrade the cell wall components of *P. infestans*. While, pathogen released pathogen-associated molecular patterns (PAMPs) were recognized by plant receptors triggering an increase in Ca^2+^ activating CaM in plant cells. Pathogen secreted polygalacturonase (PG) to degrade components of the plant cell wall, and the plant cell secreted polygalacturonase inhibitor (PGI) to block the activity of PG as a defense response. In addition, levels of CPIs were decreased, resulting in breakdown of the interactions of the proteins with Cath B, which activated the HR at local infection sites, and accumulation of enzymatic activities and H_2_O_2_-mediated callose occurred around the HR. On the other hand, numerous biotic anti-stress responses transcripts involved in plant hormone signalling, bio-stimulator, stress responses, antioxidants and ROS scavenging, were markedly induced upon Phi application. Further, photosynthetic efficiency was also maintained when spraying with Phi. The results highlighted in purple came from this study, and the events in blue from [40] and in brick-red from [59].

**Figure 8 pathogens-09-00170-f008:**
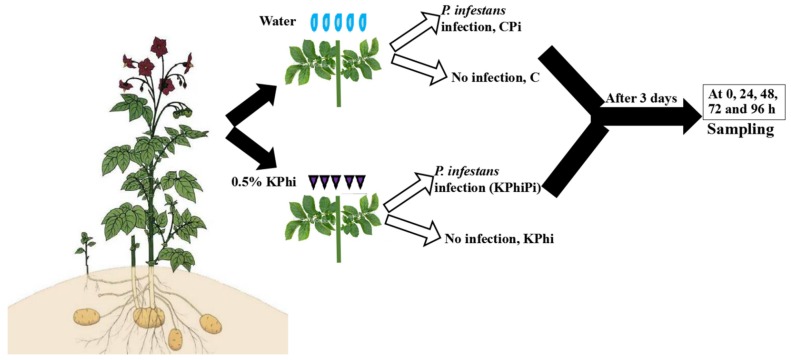
Illustration of the sampling procedure and samples taken for this work.

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
