# Peer review of "Phosphite Application Alleviates Pythophthora infestans by Modulation of Photosynthetic and Physio-Biochemical Metabolites in Potato Leaves"

_pathogens, 2020, doi:10.3390/pathogens9030170_

Round 1

Reviewer 1 Report

The manuscript entitled "Phosphite application alleviates Pythophthora infestans by modulation of photosynthetic and physio-biochemical metabolites in potato leaves" provides reliable biochemical data to explain the mechanism of the induction of resistance to the P. infestans by leaf application of potassium phosphite (KPhi). The authors have found that KPhi treatment induces the photosynthetic parameters (chlorophyll contents, carotenoids), as well as main enzymes and molecules involved in the alleviation of oxidative stress. In consequence, the treatment lowers the level of hydrogen peroxide and alleviates the lipid peroxidation. These positive effects are more profound on plants inoculated with the pathogen.

The manuscript provides essential data and adds to the knowledge about the KPhi mechanism of inducing the resistance to P. infestans in potato plants. The authors correctly planned their experiments, and the manuscript is easy to follow. From a practical point of view, the manuscript adds to the knowledge in the field. However, the level of scientific novelty presented in this manuscript is not high. The authors recently presented similar work regarding potato tubers.   Moreover, as the authors discuss, other research teams have published similar findings regarding the mechanism of induction of the resistance by KPhi to pathogens on other crops.   Most importantly, recent work by Burra et al. (2014) provides similar conclusions from transcriptome and secretome analysis. Therefore, in my personal opinion, the authors should discuss their results with this paper.   Also, the authors should try to build the model of molecular and biochemical events triggered by KPhi in plants before infection by P. infestans and happening after inoculation with the pathogens. Such a model should incorporate the data presented in the manuscript with those published in earlier works by authors and presented in the most important works of other scientific teams.

Author Response

Dear reviewer 1 of Pathogens

Thank you for the comments to our manuscript entitled: "Phosphite application alleviates Pythophthora infestans by modulation of photosynthetic and physio-biochemical metabolites in potato leaves" (Manuscript ID: pathogens-702424).

We appreciate all the comments from your review. We found the comments very helpful for improving the manuscript, and we have carefully revised the manuscript according to the suggestions. Please find our responses to the reviewers’ comments below, and all the revised parts marked in red in the revised manuscript.

Please inform us if there are any further requirements or comments. 

Sincerely,

Gefu Wang-Pruski

Professor

Department of Plant, Food, and Environmental Sciences

Faculty of Agriculture, Dalhousie University

PO Box 550, Truro, Nova Scotia, Canada B2N 5E3

Phone: (902) 893-6247

Fax: (902) 893-1404

E-mail: gefu.wang-pruski@dal.ca

Reviewer 2 Report

Mohammadi et al have explored the underlying physiological changes after phosphite application to protect potato against P. infestans. 

The major concern of the manuscript could well be language and presentations. But since it in part is so poorly presented it is not fully possible to judge whether the research is sound. With improved presentation however it could turn out that the manuscripts merits publication in MDPI Pathogens. In this review we will not point out the minor language issues that occurs through out the manuscript with maybe the exception of the introduction. Here are some major concerns on the results presented:

-the age of the plants is not clearly stated but based on the application scheme they should be more than 10 weeks old and thus be close to senescence in a greenhouse environment? If this is the case that is a major concern for the whole study.

-the scoring of disease severity is unclear. On row 110 it is even stated that Phi gives a 50% protection, but is that even possible to state with the scale te authors are using?

-there seem to be disease already at time point 0h which is not possible. In fig 1 there are very strong lesions at 72h. This is for P. infestans the stage where you possible see the first small lesions symptoms. We do not understand what is going on.

-the methods sections refers to a previous paper of the authors which this reviewer did not have access to. Even if this is the case some basic details need to be given in the paper. As for example what spore/sporangia concentration was used? 

-for all experiment it is unclear how many plants are included in each biological replicate

-it consistently says that the experiment was done in three biological protocols and was repeated twice, are both separate experiment included in the graphs? This should have led to much higher variation. What are the error bars represented in the graphs? This needs to be clearly stated.

-was some sort of normalization method used? In case not, it should probably be used as there are sometimes statistically significant differences between untreated controls, e.g. proline KPhi 0h (but there are several more). The authors should reason around this variance, we cannot see that they have done that.

-in the discussions there are several sentences which do not make sense, e.g. line 337-338. There is also an overuse of the word indicated. Does the authors themselves not trust the data?

-the conclusion sounds as this paper discovers the clear protective effect of Phi against P. infestans - this is in no way a new finding. It needs to be completely re-written.

Author Response

Dear reviewer 2 of Pathogens

Thank you for the comments to our manuscript entitled: "Phosphite application alleviates Pythophthora infestans by modulation of photosynthetic and physio-biochemical metabolites in potato leaves" (Manuscript ID: pathogens-702424).

We appreciate all the comments from your review. We found the comments very helpful for improving the manuscript, and we have carefully revised the manuscript according to the suggestions. Please find our responses to the reviewers’ comments below, and all the revised parts marked in red in the revised manuscript.

Please inform us if there are any further requirements or comments. 

Sincerely,

Gefu Wang-Pruski

Professor

Department of Plant, Food, and Environmental Sciences

Faculty of Agriculture, Dalhousie University

PO Box 550, Truro, Nova Scotia, Canada B2N 5E3

Phone: (902) 893-6247

Fax: (902) 893-1404

E-mail: gefu.wang-pruski@dal.ca

Reviewer 3 Report

Hello,

I am glad to review the manuscript titled “Phosphite Application Alleviates Pythophthora Infestans by Modulation of Photosynthetic and Physio-biochemical Metabolites in Potato Leaves”. In this study, the authors found pretreatment of Kphi can protect potato against P. infestans infection by comparison of disease severity and changes of photosynthetic parameters, as well as several physio-biochemical compounds between the control and Kphi treated potato leaves with or without P. infestans infection. The topic is very interesting and this study has produced some significant results that might be used in potato production. The experiments were well designed and the conclusion can be supported by the results. Although the logical flow and the structure of the manuscript is acceptable, the organization of individual figures from Figure 2-7 can be improved for better comparison and supporting the points. I also strongly require the improvement of the presentation of Results and Discussion before publication. My main concerns and minor concerns are listed as below.

One of my major concerns is the organization/layout of Figure 2-7. If my understanding is correct, the main point the author wanted to convey was the pretreatment of Kphi on potato leaves can protect plant defense against P. infestans infection by increasing the parameters involved in photosynthesis system, activities of antioxidant enzymes, contents of antioxidant compounds and carbohydrates, and decreasing H2O2 and MDA that are associated with ROS. Such conclusions are drawn by comparison between control and Kphi treated samples and P. infestans infection on control and Kphi treated samples. Thus, I suggest the control and Kphi treated samples at same time point are grouped together and P. infestans infection on control and Kphi treated samples at same time point are grouped together. The statistical test to calculate the P-value of significance can be conducted just between the paired-sample.

Another major concern is about the presentation of results and discussion. I think the quality of the presentation of the manuscript should be dramatically improved before publication, especiallt the Results and Discussion Sections. For example, 2.4 Effect of KPhi on non-enzymatic antioxidants in the Results section (especially the first paragraph of this part), the logical flow is not clear and presentation quality can be improved much. The 2.5 and 2.6 parts in the Results look a little bit better. There are some sentences confused me, such as, Line 42: “the multiplication of pathogens”; Line 52-54; Line 111; Line 125-127; Line 182-185; Line219-220. The quality and logic of Discussion section should be improved by putting the same points together. For example, Line 296-322 focus on photosynthetic parameters should be together and L298 of ROS should be twisted in another paragraph of L336 and L348. The last paragraph in the Introduction section for summary of this research should be improved. There are many examples of sentences and paragraphs that can be revised and improved throughout the manuscript except the Methods section (I like the Methods section). Such presentation and discussion confused readers and bury the main points in lots of noise. The first principal for presentation is the logical flow of the structure and then concise writing can help understand the main results.

Some minor concerns:

Line 83-84: Citation(s) for “previous studies”.

Line 114: Figure 1B, the statistical test conducted within paired samples makes more sense for me.

Line 122: “laves” should be “leaves”, correct?

Line 141: “P” should be italic.

Line 202: “0-96 hours. No …”, correct?

Line 504-505: Please rephase this sentence. It makes no sense for me.

Line 507: Please double check the format for references, i.e., Line 611-612; Line 706; Line 708-709, etc.

Finally, please double check spelling and typing errors throughout the manuscript.

Author Response

Dear reviewer 3 of Pathogens

Thank you for the comments to our manuscript entitled: "Phosphite application alleviates Pythophthora infestans by modulation of photosynthetic and physio-biochemical metabolites in potato leaves" (Manuscript ID: pathogens-702424).

We appreciate all the comments from your review. We found the comments very helpful for improving the manuscript, and we have carefully revised the manuscript according to the suggestions. Please find our responses to the reviewers’ comments below, and all the revised parts marked in red in the revised manuscript.

Please inform us if there are any further requirements or comments. 

Sincerely,

Gefu Wang-Pruski

Professor

Department of Plant, Food, and Environmental Sciences

Faculty of Agriculture, Dalhousie University

PO Box 550, Truro, Nova Scotia, Canada B2N 5E3

Phone: (902) 893-6247

Fax: (902) 893-1404

E-mail: gefu.wang-pruski@dal.ca

Round 2

Reviewer 1 Report

After reading the revised manuscript, I have noticed that the authors did reply to my previous comments, but their answers do not convince me. I do think that if the Authors do not wish to improve the scientific soundness of the manuscript, they should consider submission to a more potato crop oriented journal, dedicated to the practical description of control of potato diseases (like" Potato Research" for example). The manuscript is lacking a thorough discussion of the presented results with recent transcriptomic paper (which I have indicated in my previous review). Moreover, it also misses a model incorporating known data (published and presented in the manuscript) of KPhi action on a biochemical and molecular level. Therefore, with regrets, I do think that in the present form, the scientific level of the manuscript is not fitting to the standards of the Pathogens.

Author Response

After reading the revised manuscript, I have noticed that the authors did reply to my previous comments, but their answers do not convince me. I do think that if the Authors do not wish to improve the scientific soundness of the manuscript, they should consider submission to a more potato crop oriented journal, dedicated to the practical description of control of potato diseases (like" Potato Research" for example).

Response: We have done more revisions to the manuscript (see below in details) based on your comments. We believe it fits better for this journal, since our work is mostly studying the pathogen.

 The manuscript is lacking a thorough discussion of the presented results with recent transcriptomic paper (which I have indicated in my previous review).

Response: We have revised the Discussion part based on your comments. We added Figure 7 which provided a summarized model for Phi functions based on the published data and our study.Line 348.

 Moreover, it also misses a model incorporating known data (published and presented in the manuscript) of KPhi action on a biochemical and molecular level.

Response: We produced the model (Figure 7) and added detailed legend. Please review our newly submitted version.

Therefore, with regrets, I do think that in the present form, the scientific level of the manuscript is not fitting to the standards of the Pathogens.

Response: Newly revised version includes two figures (Figures 7 and 8). Figure 7 is proposed model of Phi’s functions, Figure 8 described treatments, plants sampling and inoculation. We have significantly improved the manuscript including the English. We hope this revised version is acceptable.

Reviewer 2 Report

I have read the reply of the the authors and some of them does not make sense. However some important questions have been resolved  paper around the infection. Still there is a clear need  to improve language and methods description. Potato is for example Solanum tuberosum and nothing else. Please include a figure stating when plants where samples, inoculated and analysed and whether these stages were done on whole plant or leaflets stage. One specific question, how was the samples for each biological replicate pooled, please state in methods.  

Author Response

I have read the reply of the the authors and some of them does not make sense. However some important questions have been resolved paper around the infection. Still there is a clear need to improve language and methods description. Potato is for example Solanum tuberosum and nothing else.

Response: The revised version has been edited by a native English speaker and a professional scientist in the field. All the revised are marked in red in the newly submitted version.

Please include a figure stating when plants where samples, inoculated and analysed and whether these stages were done on whole plant or leaflets stage.

Response: We produce an illustration figure (Figure 8) which indicated treatment and sampling procedures. Please see the revised version. We also revised the method section by adding some details.

One specific question, how was the samples for each biological replicate pooled, please state in methods.  

Response: In the revised manuscript, we stated in Section 4.2: “Three days after last foliar application of KPhi, a fully expanded healthy leaf from each of control or KPhi treated leaves were harvested. In total, four leaves from four different plants (C or KPhi) per replicate were detached and wrapped in aluminum foil until inoculation. A total of three replications were made for each experiment. The entire sampling procedure was illustrated in Figure 8.: We hope this explanation is satisfactory.

Reviewer 3 Report

The revised version looks much better. Authors changed the layout for the Figures with multiple comparison, which are helpful to support the main points. The text in Results and Discussion sections was largely revised. The whole quality of the manuscript has improved. 

Author Response

The revised version looks much better. Authors changed the layout for the Figures with multiple comparisons, which are helpful to support the main points. The text in Results and Discussion sections was largely revised. The whole quality of the manuscript has improved.

Response: Thanks for your positive comments. We have improve current revised version entirely and double check spelling and English requirements.

Round 3

Reviewer 1 Report

The manuscript in the current form included all my former comments. The other work concerning the phosphite induced resistance to late blight are discussed and the joint model combining know facts into a net of biochemical pathways is presented. Therefore, in my opinion, the paper can be published in the present form.

I have only one comment.

I do think that polishing the style of the entire manuscript by an experienced English native-speaker would significantly improve the manuscript reception by readers. However, I am not a native speaker myself, therefore my opinion is of minor value.